# Hierarchically porous and single Zn atom-embedded carbon molecular sieves for H$_2$ separations

Leiqing Hu[1], Won-Il Lee[2], Soumyabrata Roy[3,4], Ashwanth Subramanian[2], Kim Kisslinger[5], Lingxiang Zhu[6], Shouhong Fan[7], Sooyeon Hwang [5], Vinh T. Bui[1], Thien Tran[1], Gengyi Zhang[1], Yifu Ding[7], Pulickel M. Ajayan [3], Chang-Yong Nam [2,5] & Haiqing Lin [1] ✉

Hierarchically porous materials containing sub-nm ultramicropores with molecular sieving abilities and microcavities with high gas diffusivity may realize energy-efficient membranes for gas separations. However, rationally designing and constructing such pores into large-area membranes enabling efficient H$_2$ separations remains challenging. Here, we report the synthesis and utilization of hybrid carbon molecular sieve membranes with well-controlled nano- and micro-pores and single zinc atoms and clusters well-dispersed inside the nanopores via the carbonization of supramolecular mixed matrix materials containing amorphous and crystalline zeolitic imidazolate frameworks. Carbonization temperature is used to fine-tune pore sizes, achieving ultrahigh selectivity for H$_2$/CO$_2$ (130), H$_2$/CH$_4$ (2900), H$_2$/N$_2$ (880), and H$_2$/C$_2$H$_6$ (7900) with stability against water vapor and physical aging during a continuous 120-h test.

Hydrogen (H$_2$) has been touted as a clean energy carrier and a key pillar for a carbon-neutral or negative society, and its production and transportation have attracted significant interest in lowering the cost. In particular, H$_2$ purification and recovery can be costly and energy-intensive, such as H$_2$/CO$_2$ separation for blue H$_2$ production from fossil fuels or biomasses[1,2], H$_2$/CH$_4$ separation for co-delivery with CH$_4$ using natural gas pipelines, and H$_2$/hydrocarbon separations for its recovery from refinery streams[3]. Membrane technology features high energy efficiency, excellent scalability, and an absence of waste emissions[4–6]. H$_2$ is less condensable than other gases, as indicated by its lower critical temperature, and thus, it has unfavorable solubility selectivity. Therefore, for H$_2$ separations, membrane materials should have a strong size-sieving ability because H$_2$ (with a kinetic diameter of 2.89 Å) is smaller than other gases, such as CO$_2$ (3.3 Å), N$_2$ (3.64 Å), CH$_4$ (3.8 Å), and C$_2$H$_6$ (4.44 Å). State-of-the-art membranes are often based on amorphous polymers with randomly distributed and ill-controlled

free volumes, leading to low H$_2$/CO$_2$ selectivity[7]. Polymers can also be cross-linked to reduce the free volume sizes and increase size-sieving ability, which is usually accompanied by decreased H$_2$ permeability[7,8]. Materials with well-controlled sub-nanometer pores were also investigated for these applications, such as graphene oxides (GO)[9], metal-organic frameworks (MOFs)[10,11], and MXene[12]. However, fabricating these membranes on a commercial scale is challenging.

Nanoporous carbons with highly developed micropores and mesopores have been extensively explored for energy storage applications, adsorbents, and membranes for gas separations[2,5,13] and liquid separations[14]. Unlike MOFs with uniform pore or channel sizes, carbon molecular sieving (CMS) membranes can be facilely synthesized by pyrolysis of polymer precursors, and they comprise multi-modal pores including ultramicropores (<7 Å) or bottlenecks precisely sieving penetrant molecules and microcavities (7–20 Å) promoting molecular permeation[15–17]. As such, tailoring CMS porous structures presents a

[1]Department of Chemical and Biological Engineering, University at Buffalo, The State University of New York, Buffalo, NY, USA. [2]Department of Materials Science and Chemical Engineering, Stony Brook University, Stony Brook, NY, USA. [3]Department of Materials Science and NanoEngineering, Rice University, Houston, TX, USA. [4]Department of Sustainable Energy Engineering, Indian Institute of Technology Kanpur, Kanpur, Uttar Pradesh, India. [5]Center for Functional Nanomaterials, Brookhaven National Laboratory, Upton, NY, USA. [6]Department of Energy, National Energy Technology Laboratory, Pittsburgh, PA, USA. [7]Department of Mechanical Engineering, University of Colorado, Boulder, CO, USA. ✉e-mail: haiqingl@buffalo.edu

great potential to simultaneously improve $H_2$ permeability and $H_2$/gas selectivity, overcoming the permeability/selectivity trade-off that confines most conventional gas separation membranes[4].

Pore structures of CMS membranes depend on precursor structures[2,5,18], carbonization temperature ($T_c$)[19], and carbonization atmosphere[20,21]. The ultramicropore sizes can be reduced to precisely separate $H_2$ from other gases, which, however, lowers the $H_2$ permeability[5,13,20,22]. On the other hand, mixed matrix materials (MMMs) containing inorganic nanofillers dispersed in polymers provide a versatile materials platform as the CMS precursors[23], as both polymers and fillers can be orthogonally fine-tuned to improve the separation performance of $CO_2/CH_4$[24], $CO_2/N_2$[25], and olefin/paraffin[26,27]. However, their ultramicropore sizes are 5–7 Å due to poor compatibilities and disturbed packings of MMM precursors, thus leading to the insufficient molecular sieving ability for $H_2$/gas separations.

Here, we report a distinct series of hybrid CMS materials derived from a supramolecular MMM (sMMM) containing zeolitic imidazolate framework-8 (ZIF-8) in polybenzimidazole (PBI) synthesized by an in-situ growth method (Fig. 1a). The benzimidazoles in PBI are similar to the ligands (2-methylimidazole or 2-mIm) in forming ZIF-8, and thus, the polymer chains are uniquely incorporated in the ZIF-8, forming amorphous ZIF-8 with strong size-sieving ability in homogeneous MMMs[28]. For instance, an sMMM containing amorphous ZIF-8 (11 mass%) and crystalline ZIF-8 (9.1 mass%) was synthesized as the CMS precursor (Supplementary Table 1), and it exhibits $H_2/CO_2$ selectivity (29) higher than PBI (17) because of the strong size sieving ability of the amorphous ZIF-8, despite the low $H_2/CO_2$ selectivity in the crystalline ZIF-8. By contrast, an MMM comprising 10 mass% crystalline ZIF-8 and PBI

prepared by physical blending exhibits $H_2/CO_2$ selectivity of only 8.1[28]. We carbonized this sMMM at $T_c$ values of 400–900 °C (named sMMM$T_c$) and demonstrate their suitable polymodal free volumes with superior $H_2$/gas separation properties. More importantly, the carbonization of amorphous ZIF-8 leads to single Zn atom (2.6 Å) and nanoclusters, which are well dispersed inside the porous carbons and have the right sizes to fine-tune the pore sizes for improved $H_2$/gas separation properties. The sMMM CMS membranes exhibit $H_2/CO_2$ selectivity up to 130, $H_2/CH_4$ selectivity up to 2900, and $H_2/C_2H_6$ up to 7900, superior to the leading polymeric materials and far surpassing Robeson's 2008 upper bounds. The combination of amorphous MOFs and carbonization provides an effective approach to designing tunable hierarchical pores in carbon materials facilitated by single atoms and clusters.

## Results

### Materials synthesis and characterization

We investigate the chemical and morphological structures of the sMMM CMS materials. The carbonization changes the sMMM films from orange to dark because of the loss of organic groups (Supplementary Fig. 1a), and increasing $T_c$ makes the films more brittle and rigid (Supplementary Fig. 1b, c). Figure 1b compares the wide-angle x-ray diffraction (WAXD) patterns of the CMS films at various $T_c$ values. Both sMMM and sMMM400 exhibit characteristic peaks of crystalline ZIF-8 at 8° and 13°[28], which disappear at $T_c$ of 500 °C or above, indicating the destruction of crystalline ZIF-8. Carbonization increases the d-spacing (the average intersegmental distance among polymer chains calculated using Bragg's equation). On the other hand, increasing $T_c$ consistently decreases the d-spacing in the carbonized membranes,

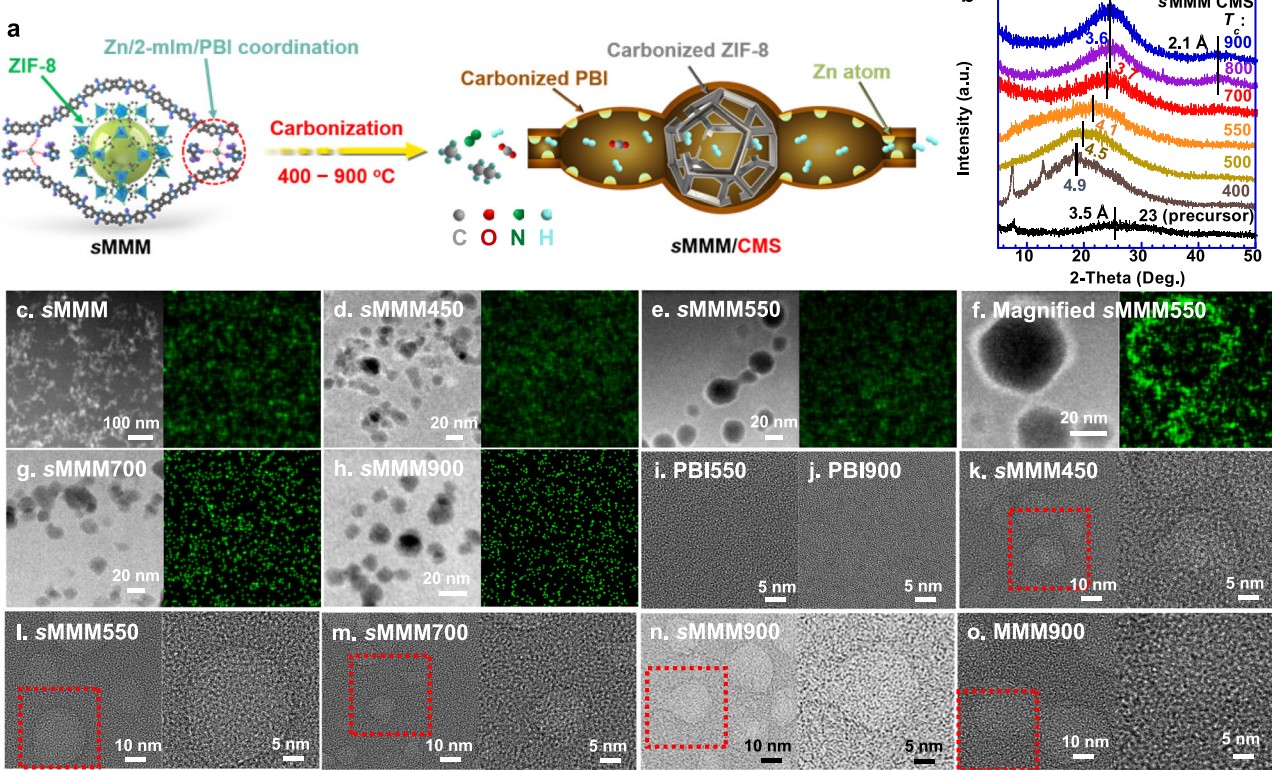

**Fig. 1 | Morphology of hierarchically porous carbon molecular sieve (CMS) membranes with single atoms and clusters. a** Schematic illustration of carbonization of supramolecular mixed matrix material (sMMM) to form the CMS material containing polymodal free volumes from 400 to 900 °C. **b** Wide-angle x-ray diffraction (WAXD) patterns showing the effect of $T_c$ on d-spacing values. High-angle annular dark field scanning transmission electron microscopy (HAADF-STEM) (left) and STEM-energy dispersive spectroscopy (EDS) Zn (green spots) elemental maps (right) of (**c**) sMMM, (**d**) sMMM450, (**e**) sMMM550, (**f**) magnified sMMM550, (**g**) sMMM700, and (**h**) sMMM900. Bright-field TEM images of (**i**) PBI550, (**j**) PBI900, (**k**) sMMM450, (**l**) sMMM550, (**m**) sMMM700, (**n**) sMMM900, and (**o**) MMM900. In **k**–**o**, the right images are the magnified region (red square) of the left image. In **a**, gray, blue, and purple circles on PBI chains represent C, N, and H atoms, respectively.

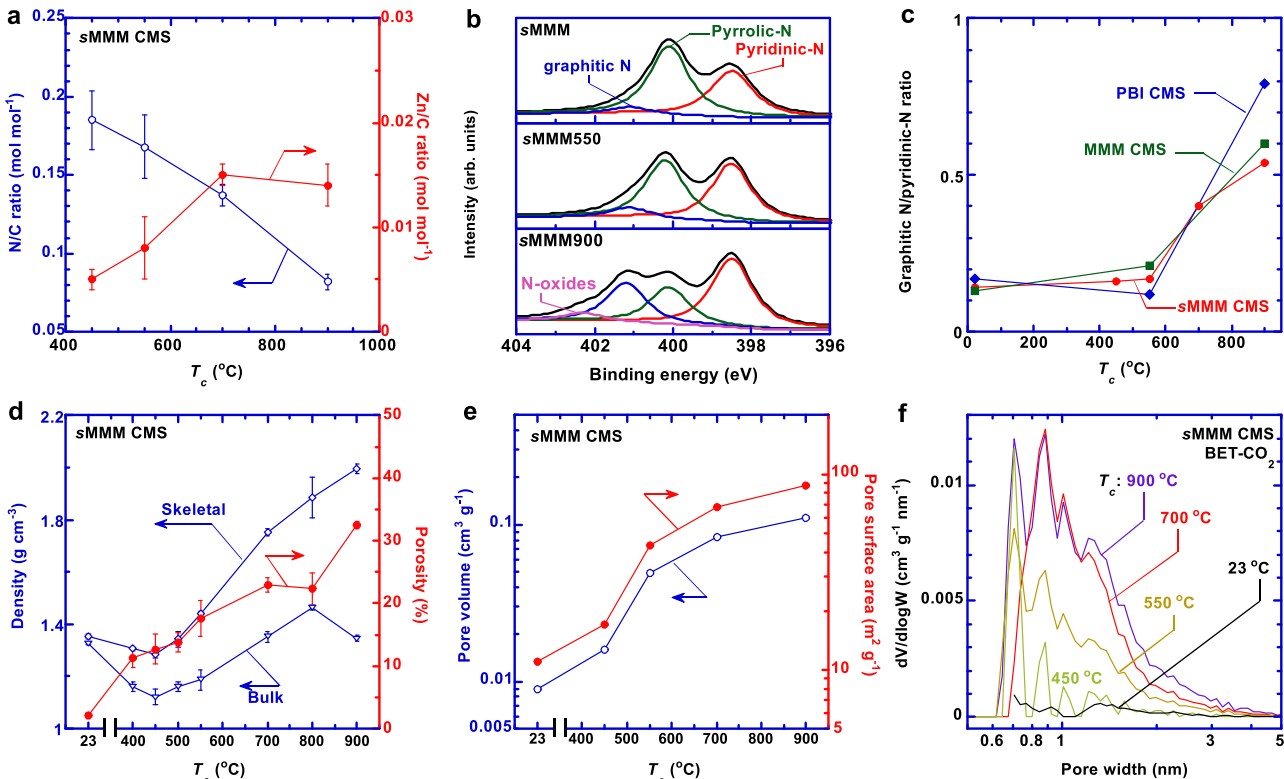

**Fig. 2 | Chemistry and pore structures of sMMM CMS membranes.** Effect of the carbonization temperature ($T_c$) on (**a**) the N/C molar ratio, (**b**) N 1$s$ spectra, and (**c**) the ratio of graphitic N to pyridinic-N determined by x-ray photoelectron spectroscopy (XPS). **d** Skeletal density, bulk density, and porosity. **e** Pore volume and pore surface area, and (**f**) pore size distributions obtained by non-local density functional theory (NLDFT) from $CO_2$ sorption at 0 °C. Note: $T_c = 23$ °C in Fig. **c–f** represents the sMMM precursor. Error bars in Fig. **a**, **d** are standard deviations from 3 samples.

indicating an increased molecular size-sieving ability. Carbonization at 800 °C or above results in a new peak at 44°, corresponding to a $d$-spacing of 2.1 Å, characteristics of graphite planes[19].

Figure 1c–h displays the high-angle annular dark field scanning transmission electron microscopy (HAADF-STEM) images of CMS films. ZIF-8 nanoparticles with uniform sizes of 20–40 nm are well dispersed without interfacial voids in the sMMM, confirmed by its scanning electron microscopy (SEM) images (Supplementary Figs. 2a and 3a). The STEM-energy dispersive spectroscopy (EDS) elemental mapping demonstrates a uniform distribution of Zn element because of the homogenous structure of the sMMM containing amorphous ZIF-8 (Supplementary Figs. 2 and 3)[28]. However, both sMMM450 and sMMM550 exhibit a 1–2 nm thick interfacial region with strong contrast in HAADF-STEM images (Fig. 1d–f), indicating agglomerations of elements with a high atomic number ($Z$) because the image contrast is proportional to $Z^2$. STEM-EDS elemental map confirms that the strong contrast in HAADF-STEM images is indeed due to the segregation of Zn derived from the ZIF-8 degradation. However, the Zn agglomerates gradually disappear in sMMM700 and sMMM900 because of the diffusion of Zn elements and their dispersion in the carbon structures (Fig. 1g, h). Similar trends are also observed from the SEM images, where bright Zn agglomerates appear for sMMM550 and disappear for sMMM900 (Supplementary Fig. 2).

Bright-field TEM was also used to monitor the evolution of carbon structures in the CMS films. PBI550 (Fig. 1i) shows a homogeneous structure, while PBI900 (Fig. 1j) displays graphitic structures consistent with their Raman spectra (Supplementary Fig. 4). By contrast, the CMS films exhibit a clear contrast between the carbonized PBI and ZIF-8 (Fig. 1k–o), indicating different carbon nanostructure phases. The dark boundary reflects the Zn elements, and it becomes weaker with increasing $T_c$ because of the diffusion of the Zn atoms to the carbon structures, consistent with the HAADF-STEM images.

Increasing $T_c$ also decreases the $I_D/I_G$ ratio in Raman spectra (Supplementary Fig. 4a), indicating the increased graphitic structure and consistent with the enhanced WAXD peak at 44° (Fig. 1b). Notably, because of the lack of oxygen-containing groups in PBI, there are no crystalline ZnO lattices in these CMS samples, as indicated by the absence of their characteristic peaks (30°–40°) in the WAXD patterns (Fig. 1b and Supplementary Fig. 4b)[25,26].

Figure 2a–c and Supplementary Fig. S5 exhibit the elemental analysis of sMMM CMS films using x-ray photoelectron spectroscopy (XPS). The N/C molar ratio decreases with increasing $T_c$, resulting from the loss of N-containing groups in the order of the 2-mIm from the amorphous ZIF-8 (300 °C), then ZIF-8 (450 °C), and finally imidazole rings on the PBI chains (480 °C)[2,28,29]. By contrast, increasing $T_c$ increases the Zn/C molar ratio (because of the loss of organic carbons) before leveling off at 700 °C, indicating that the Zn element is retained even at 900 °C. This behavior vastly differs from the carbonized ZIF-8 at 900 °C (where most Zn element is removed[30]) probably because of the restricted Zn diffusion by PBI. The N 1$s$ peaks in sMMM and sMMM550 can be deconvoluted to pyridinic, pyrrolic, and graphitic nitrogens[30], corresponding to binding energies at 398.5, 400.1, and 401.1 eV, respectively (Fig. 2b). Increasing $T_c$ enhances the graphitic N peak and its ratio to pyridinic N (Fig. 2c). Carbonization at 900 °C introduces a new N-oxide peak at 402.3 eV for all three samples (Supplementary Fig. 5).

The skeletal density ($\rho_s$) of the CMS samples was determined using a gas pycnometer, and their bulk density ($\rho_b$) was calculated as the measured mass/volume ratio. Increasing $T_c$ initially decreases $\rho_s$ and $\rho_b$ due to the degradation of amorphous ZIF-8 before increasing (Fig. 2d) because of the generated densified carbon structures[2]. The porosity ($\varepsilon = 1 - \rho_b/\rho_s$) increases with increasing $T_c$ due to the increased mass loss (Supplementary Fig. 6a). For example, carbonization at 900 °C increases the $\varepsilon$ value from 0 to 33%, coherent with the mass loss

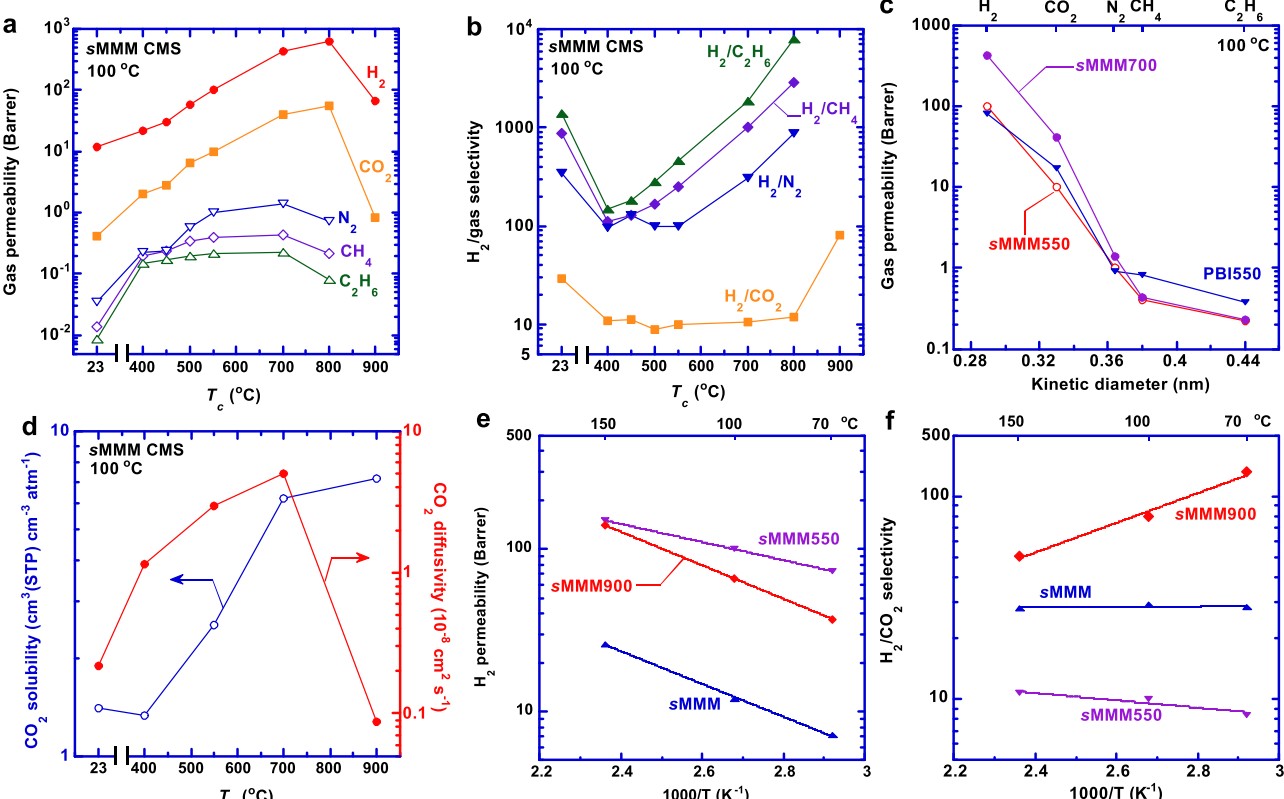

**Fig. 3 | Pure-gas transport characteristics of *s*MMM CMS films with a thickness of ≈35 μm and an effective area of 0.5–1 cm².** **a** Gas permeability and (**b**) H₂/gas selectivity at 100 °C showcasing the flexible tunability of separation properties by varying carbonization temperature (*T*c). **c** Pure-gas permeability of *s*MMM700, PBI550, and MMM550 at 100 °C illustrating molecular size cutoff of 2.89–3.64 Å. **d** Effect of *T*c on CO₂ solubility and diffusivity at 100 °C. Temperature-dependent behavior of (**e**) H₂ permeability and (**f**) H₂/CO₂ selectivity described using the Arrhenius equation.

of 28%. We further analyze the pore structures of CMS films based on their CO₂ adsorption isotherms at 0 °C (Supplementary Fig. 6b,c). Increasing $T_c$ increases the pore volume and surface area (Fig. 2e), consistent with the increased porosity. The *s*MMM CMS films display pore volumes (<0.1 cm³/g in Supplementary Table 1) lower than other CMS materials derived from PIM-1, cellulose, and 6FDA-based polyimides[13,20,27], due to the rigid packing of the *s*MMM precursor. The *s*MMM900 exhibits higher intensity of the ultramicropores (0.6 – 0.8 nm) than other samples (Fig. 2f). The trends are similar to the pore size distribution determined using N₂ at −196 °C (Supplementary Table 2 and Fig. 6d–h) and the *d*-spacing results. Both CO₂ (0 °C) and N₂ sorption (−196 °C) lead to similar values of pore volumes and surface areas, which has been observed for carbon materials with carbonization-derived mass losses of 5–30%[31]. Notably, both sorption measurements do not show ultramicropores of 3–5 Å because N₂ is too large to access the ultramicropores[5,20], and our apparatus cannot obtain stable CO₂ pressure ranges below 10⁻⁵ to accurately evaluate ultramicropores. The simultaneously increased porosity and decreased *d*-spacing (or increased intensity for ultramicropores) indicate that higher $T_c$ creates more but smaller ultramicropores, leading to the multimodal pore size distribution.

**Pure-gas separation properties and mechanisms**
We determine pure-gas permeability of the *s*MMM CMS membranes at 100 °C to probe the pore sizes of the carbon structures (Fig. 3a and Supplementary Table 3). Gas permeability decreases with increasing molecular size in the following order, H₂, CO₂, N₂, CH₄, and C₂H₆. The permeability of N₂, CH₄, and C₂H₆ increases as $T_c$ increases to 700 °C (because of the increased porosity) before decreasing (due to the decreased *d*-spacing as shown in Fig. 1b). H₂/CH₄ and H₂/C₂H₆

selectivity decreases when carbonized at 400 °C (because of the increased *d*-spacing) before dramatically increasing (Fig. 3b), suggesting the formation of ultramicropores between 2.89 and 3.8 Å. In contrast, H₂/N₂ selectivity does not increase until $T_c$ reaches 550 °C, indicating that the ultramicropores decrease to 3.64 Å at $T_c$ greater than 550 °C. The *s*MMM800 exhibits H₂/N₂, H₂/CH₄, CO₂/CH₄, and CO₂/C₂H₆ separation properties surpassing their corresponding upper bounds (Supplementary Fig. 7)[32]. Increasing $T_c$ decreases H₂/CO₂ selectivity until $T_c$ reaches 800 °C when the bottleneck decreases to 3.3 Å. Particularly, *s*MMM900 exhibits H₂/CO₂ selectivity of 80, one of the highest values reported in the literature.

Figure 3c presents the effect of the penetrant size on gas permeability in *s*MMM700, PBI550, and MMM550. All three CMS membranes show a molecular cut-off between 3.3 and 3.64 Å. Gas transport in CMS materials is usually described using the solution-diffusion model, where permeability is decoupled into gas solubility and diffusivity[5,18]. CO₂ and C₂H₆ sorption isotherms were determined at 100 °C (Supplementary Fig. 8). Due to its low condensability, H₂ sorption is below the detection limit of our sorption apparatus, and C₂H₆ can be used as a surrogate for H₂[33]. Figure 3d shows that CO₂ solubility increases with increasing $T_c$ due to the increased porosity (Supplementary Table 4), and CO₂ diffusivity (calculated by permeability divided by solubility) follows the same trend as permeability, i.e., it increases before decreasing.

We investigate pure-gas H₂/CO₂ separation behaviors of *s*MMM, *s*MMM550, and *s*MMM900 at 70–150 °C, an interesting temperature range for syngas processing[7]. Both H₂ and CO₂ permeability increase with increasing temperature, and they can be satisfactorily described using the Arrhenius equation (Fig. 3e)[34]. Compared to *s*MMM and *s*MMM900, *s*MMM550 has the lowest value of activation energy ($E_{P,A}$)

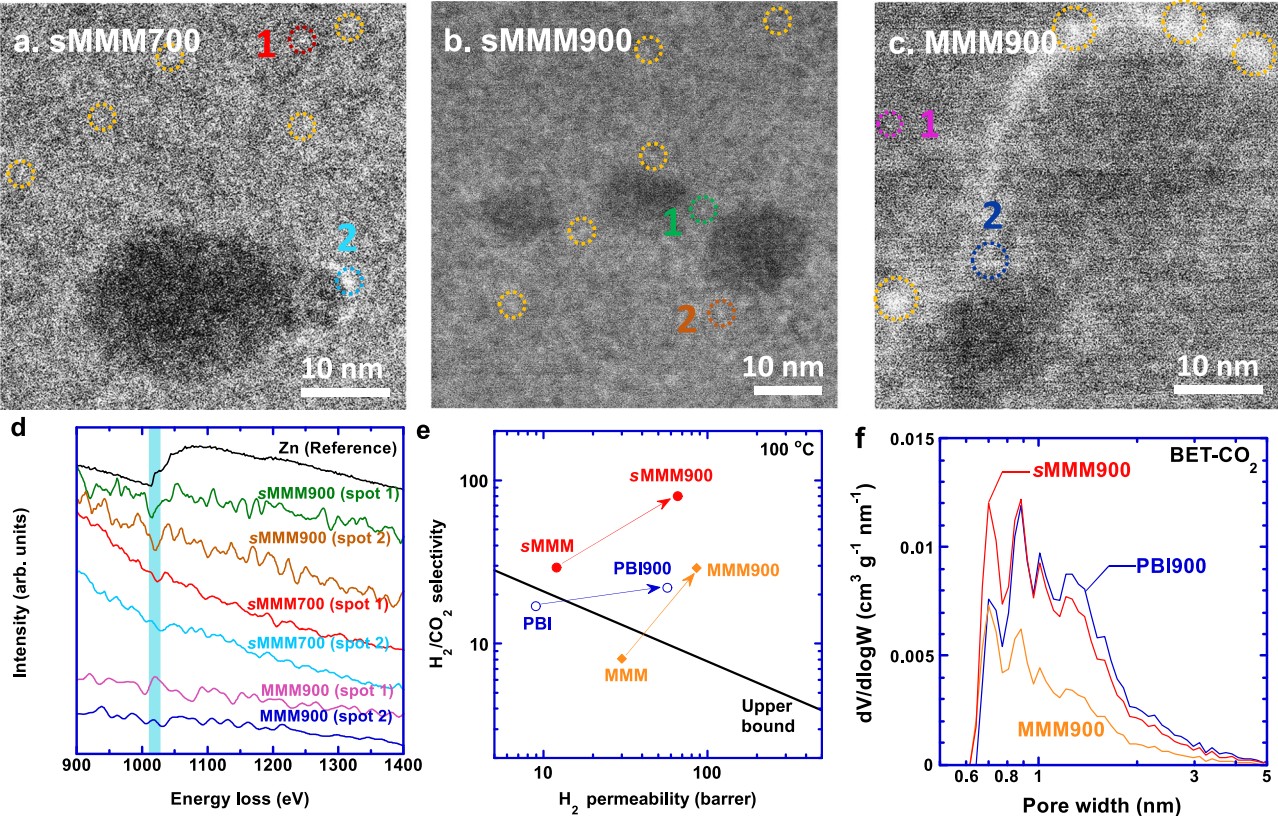

**Fig. 4 | Single Zn atoms and clusters to improve H₂/CO₂ separation properties in sMMM CMS membranes.** HAADF-STEM images of (**a**) *s*MMM700, (**b**) *s*MMM900, and (**c**) MMM900. The numbers and circles indicate the single Zn atom or Zn atom clusters, and they are shown in Fig. d correspondingly. **d** Electron energy loss spectra (EELS) of *s*MMM700, *s*MMM900, and MMM900 referenced with Zn L edge of Zn metal. **e** Superior H₂/CO₂ separation properties in sMMM900. **f** Pore size distributions obtained by NLDFT from CO₂ adsorption at 0 °C.

for H₂ and CO₂ permeation (Supplementary Tables 6 and 7) because of the combination of high porosity and large ultramicropores; *s*MMM900 exhibits the highest $E_{P,A}$ value for H₂ and CO₂ permeation among all samples because of its lowest *d*-spacing and smallest ultramicropores. *s*MMM900 exhibits the most negative value of ($E_{P,H_2} - E_{P,CO_2}$) among all sMMM CMS samples, leading to decreased H₂/CO₂ selectivity with increasing temperature (Fig. 3f). This behavior is consistent with the strongest size-sieving ability in *s*MMM900. Similarly, MMM900 exhibits H₂/CO₂ selectivity decreasing with increasing temperature and a negative value of ($E_{P,H_2} - E_{P,CO_2}$) while MMM and MMM550 display the selectivity independent of temperature because of the weak size-sieving ability (Supplementary Fig. 9)[2]. Noticeably, *s*MMM900 exhibits H₂/CO₂ selectivity of 130 at 70 °C, the highest value obtained for our CMS materials in this study.

We wish to show here that the carbonization of sMMM forms single Zn atoms and clusters, which are well dispersed in the carbon structure and can further tune the sub-nanopores and improve H₂/CO₂ separation properties. Figure 4a–c shows the formed single Zn atoms and clusters in the HAADF-STEM images (as shown in circles) for *s*MMM700, *s*MMM900, and MMM900, respectively. The MMM900 still exhibits a boundary formed by Zn aggregates because of the stability of the crystalline ZIF-8. Both *s*MMM700 and *s*MMM900 show a more uniform distribution of single atoms and clusters than MMM900 because the amorphous ZIF-8 in *s*MMM is easier to degrade for the Zn atoms to diffuse to form single atoms or clusters than the crystalline ZIF-8 in the MMM. Figure 4d presents Zn L edge electron energy-loss spectra (EELS) of *s*MMM700, *s*MMM900, and MMM900. The onset of the Zn *L3* edge is 1020 eV, which is presented in the spectra of *s*MMM700 and *s*MMM900, further confirming the formation of the single atoms and clusters. However, the Zn *L3* edge is not visible for MMM900, presumably because of the few atoms selected for the spots.

Figure 4e shows that *s*MMM900 displays higher H₂/CO₂ selectivity than PBI900 and MMM900, and its separation performance is well above Robeson's upper bound[33,35] because of its suitable pore size and distribution (Fig. 4f). The *s*MMM900 exhibits narrower and more intensive peaks for the ultramicropores than PBI900 and MMM900 based on the CO₂ adsorption curves at 0 °C (Fig. 4f), which can be partially attributed to the precursor with the lowest *d*-spacing (Fig. 1b and Supplementary Fig. 4b) and, importantly, the single Zn atoms and clusters that diffuse in the nanopores and augment their size and distribution in *s*MMM900.

The *s*MMM900 shows higher H₂ permeability than PBI900 despite its lower porosity (Fig. 4e). On the other hand, MMM900 has the highest H₂ permeability among the three samples because of its lowest bulk density and highest porosity (50%) (Supplementary Fig. 6i,j and Table 2), compared to *s*MMM900 (33%) and PBI900 (36%), probably because the carbonized crystalline ZIF-8 forms rigid structures disrupting the stacking of the graphitic domains[36].

## Mixed-gas separation performance

We further evaluate *s*MMM900 with H₂/CO₂ gas mixtures at different temperatures due to its superior pure-gas H₂/CO₂ separation properties. Both mixed-gas H₂ and CO₂ permeability decrease slightly with increased CO₂ partial pressure at 100 °C (Fig. 5a), owing to the competitive sorption that commonly occurs in porous materials[2,28]. As such, a significant decrease can be observed from pure-gas H₂/CO₂ selectivity of 80 to mixed-gas H₂/CO₂ selectivity of ≈60 (Fig. 5b). Mixed-gas H₂/CO₂ selectivity is almost independent of CO₂ partial pressure, indicating the absence of CO₂ plasticization.

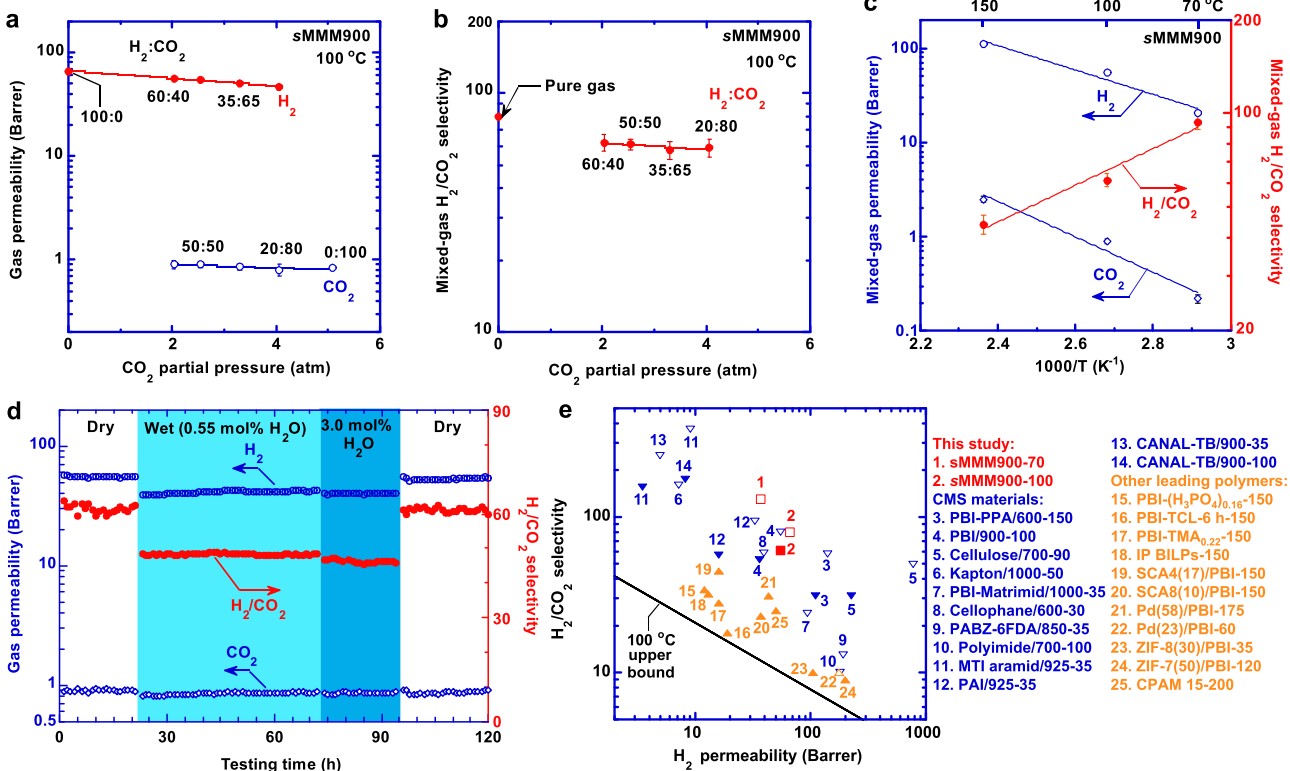

**Fig. 5 | Superior and stable mixed-gas $H_2/CO_2$ separation properties of *s*MMM900 with a thickness of ≈35 μm and an effective area of 0.5–1 cm² at 100 °C. a** Mixed-gas $H_2$ and $CO_2$ permeability and (**b**) $H_2/CO_2$ selectivity as a function of $CO_2$ partial pressure at 5 atm. **c** Mixed-gas $H_2/CO_2$ separation property as a function of the temperature at 5 atm with $H_2/CO_2$ 50/50. Error bars in Fig. a–c are standard deviations from more than 3 measurements. **d** Long-term stability in varied humidity values for 120 h ($H_2/CO_2$ 50/50, 5 atm). **e** Comparison with state-of-the-art membrane materials, including CMS materials (PBI-PPA/600, PBI/900,

Cellulose/700, Kapton/1000, PBI-Matrimid/1000, Cellophane/600, PABZ-6FDA/850, polyimide/700, MTI aramide/925), and leading polymeric materials (PBI-$(H_3PO_4)_{0.16}$, PBI-TCL-6 h, PBI-TMA$_{0.22}$, IP BILPs, SCA4/PBI-17, SCA8/PBI-10, Pd/PBI-58, Pd/PBI-23, ZIF-8/PBI-30, ZIF-7/PBI-50, CPAM-15). For CMSy-z, y and z represent $T_c$ and testing temperature (°C), respectively. For materials 11-21, the numbers after the dash represent testing temperature (°C). The open and filled symbols indicate mean pure-gas and mixed-gas ($H_2/CO_2$: 50/50) performance, respectively (Supplementary Table 8).

Figure 5c shows that mixed-gas $H_2$ and $CO_2$ permeability from 70 to 150 °C can be described by the Arrhenius equation. The $E_{P,A}$ values (Supplementary Table 7) for both gases are higher than those of pure-gas permeation due to the competitive sorption[28]. For example, $E_{P,A}$ values of pure-gas and mixed-gas $H_2$ permeation are 20 ± 1 and 25 ± 2 kJ/mol, respectively.

Figure 5d demonstrates the long-term stability of *s*MMM900 when challenged with simulated syngas containing different water vapor contents at 100 °C and 4 atm. The film exhibits stable $H_2/CO_2$ selectivity of ≈60 and $H_2$ permeability of ≈55 Barrer during the first 21 h of dry condition. Introducing 0.55 mol% water vapor into the feed decreases $H_2$ permeability by 23% and $H_2/CO_2$ selectivity to 48 because the adsorbed water blocks the pores. On the other hand, an increase of water vapor content to 3 mol% has a negligible effect on $H_2/CO_2$ separation properties. After switching back to the dry binary mixture, both $H_2$ permeability and $H_2/CO_2$ selectivity have recovered to the initial values, demonstrating its stability against physical aging and water vapor during this 120-h test. Moreover, the CMS film shows consistent $H_2/CO_2$ separation performance for 20 days, indicating good stability.

Figure 5e compares $H_2/CO_2$ separation performance of *s*MMM900 with state-of-the-art CMS and polymeric materials. Most CMS materials exhibit $H_2/CO_2$ separation properties superior to polymers and surpassing their upper bound at 100 °C due to the bimodal porous structure. Notably, the upper bound was estimated from Robeson's one at 35 °C using an activated diffusion model[7,33]. Nevertheless, the *s*MMM900 with hierarchical porous structures displays one of the best combinations of $H_2$ permeability and $H_2/CO_2$ selectivity among CMS materials.

## Discussion

We develop a series of CMS membranes by carbonization of supramolecular mixed matrix materials (*s*MMM) containing amorphous and crystalline ZIF-8. Carbonization temperature can be used to engineer hierarchical nano- and micro-pores, including the bottlenecks of 3.3–3.8 Å, realizing superior $H_2$/gas separation performance for the production, transportation, and recovery of $H_2$ as a clean energy carrier. The sMMM CMS membranes exhibit $H_2/CO_2$ selectivity up to 130, $H_2/CH_4$ selectivity up to 2900, and $H_2/C_2H_6$ up to 7900, superior to the leading CMS and polymeric materials and far surpassing Robeson's 2008 upped bounds. Our approach of CMS membranes doped by single atoms and nanoclusters derived from amorphous MOFs (with great versatility) can effectively fine-tune porous structures and holds potential for applications involved with molecular diffusion, such as membranes, adsorption, catalysts, and energy storage.

We expect that the sMMM CMS films can be further optimized using a variety of parameters to achieve a desirable combination of separation properties and mechanical strengths, including ZIF-8 concentration, MOF types, carbonization temperature, etc. Future work must also develop these materials into industrial membranes with thin selective layers to obtain high gas permeance.

## Methods
### Materials
Celazole PBI powder was obtained from PBI Performance Product Inc. (Charlotte, NC). Zinc nitrate hexahydrate ($Zn(NO_3)_2 \cdot 6H_2O$), 2-methylimidazole ($C_4H_6N_2$, 2-mIm), and N, N-dimethylformamide (DMF) were acquired from Sigma-Aldrich Corporation (St. Louis, MO).

Methanol (99.8%, MeOH) was obtained from Thermo Fisher Scientific (Waltham, MA). Gas cylinders of $N_2$, $H_2$, $CO_2$, $CH_4$, and $C_2H_6$ with ultrahigh purity were acquired from Airgas Inc. (Buffalo, NY).

## Preparation of sMMM, PBI, and CMS films

The sMMM films were prepared using the following steps[28]. First, 10 g PBI powders were added to 90 g DMF, and the solution was heated at 160 °C for 12 h to allow the PBI to dissolve. After filtering through a 1-µm glass fiber filter, a solution with ~ 6 wt% PBI concentration was obtained and then diluted to 3 wt% by adding more DMF. Second, $Zn(NO_3)_2{\cdot}6H_2O$ and 2-mIm were dried in a vacuum oven at 200 °C. Then, 2-mIm (110 mg) was added to a PBI solution (6 g), and $Zn(NO_3)_2$ (50 mg) was dissolved in DMF (3 g). The latter solution was added dropwise in the 2-mIm/PBI/DMF solution and then sonicated for 10 min. Third, the mixed solution was poured into a glass petri dish and dried at 60 °C overnight to obtain a solid film. The film was further dried under vacuum at 150 °C for 72 h and then immersed in fresh methanol for 24 h to remove the residual DMF. Finally, the film with an average thickness of ≈35 µm was dried in a vacuum oven at 100 °C for later use. Freestanding films of PBI and MMMs (~35 µm) were prepared by the solution casting method.

To prepare a CMS film, an sMMM or PBI film (with a mass of $m_0$) was sandwiched between two nonporous alumina ceramic sheets (4.5 inches × 4.5 inches × 0.025 inches, McMaster Carr, IL). The assembly was placed in a tube furnace with a $N_2$ flow of 200 mL/min (Supplementary Fig. 10). The temperature was ramped up from 20 °C to the carbonization temperature ($T_c$) at 10 °C /min and then kept at $T_c$ for 2 h. After that, the furnace was cooled down naturally with the $N_2$ flow. The mass ($m_1$) of the obtained CMS film was measured. The mass loss ($L_m$, %) by carbonization can be calculated using the following equation:

$$L_m = (m_0 - m_1)/m_0 \times 100\% \qquad (1)$$

The carbonization barely changed the film thickness (≈35 µm).

## Characterizations of CMS films

A vertex 70 Bruker spectrometer (Billerica, MA) was used for the FTIR measurement. WAXD patterns were obtained using a Rigaku Ultima IV X-ray diffractometer (Rigaku Analytical Devices, Wilmington, MA). A TG209 F1 Iris® Netzch TGA was used to simulate sample carbonizations. The film bulk density was calculated from the measured mass and volume, and the skeletal density was obtained using a Gas Pycnometer (Micromeritics Instrument Corporation, Norcross, GA). The stress-strain curve was obtained by applying uniaxial tensile loading on the sample at an initial strain of 0.1% and 1.0%/min until the sample fractured on a DMA (Q800 TA Instrument). A STEM (FEI Talos F200X; 200 kV; equipped with the EDS elemental mapping capability) was used to detect morphologies of samples. Hitachi HD2700C dedicated STEM at an accelerating voltage of 200 kV, equipped with a Cs probe corrector and Gatan Enfinium electron energy loss spectroscope, was utilized for investigating Zn atoms and clusters. XPS was performed with a PHI5000 VersaProbe III scanning XPS probe (Physical Electronics Inc., Chanhassen, MN, USA) equipped with a monochromated aluminum kα radiation source. Each XPS spectrum was collected over a sample area of 100 µm in diameter, and 3 sample areas were examined for each specimen. XPS spectra were calibrated by setting adventitious C 1s binding energy at 284.8 eV. Atomic concentrations were calculated from the obtained XPS spectra using the CasaXPS package and manufacturer-provided sensitivity factors. $CO_2$ and $N_2$ adsorption and desorption isotherms were collected at 273 and 77 K, respectively, using a Quantachrome Autosorb-iQ3-MP/Kr BET Surface Analyzer. The samples were outgassed at 120 °C for 12 h under a vacuum before each measurement. For both $N_2$ and $CO_2$ isotherm measurements, 67 adsorption and 40 desorption points were collected. Surface areas of samples were determined via the BET method by fitting gas adsorption points between the pressure range 0.05–0.3 bar. NLDFT model was used to obtain the pore size distribution from gas adsorption isotherms.

Pure-gas permeability was determined using a constant-volume and variable-pressure apparatus for CMS samples with an effective area of 0.5–1 cm². Three samples were tested for each CMS membrane, and all measurements were taken after the steady state was reached. Pure-gas sorption isotherms of $CO_2$ and $C_2H_6$ were determined using a gravimetric sorption analyzer of IGA 001 (Hiden Isochema Ltd., Warrington, UK). The uncertainty for gas permeability and solubility is estimated to be <10% using an error propagation method[37]. Mixed-gas permeability was determined using a constant-pressure and variable-volume apparatus. Feed gas mixtures of $H_2$ and $CO_2$ were prepared by in-line mixing, and $N_2$ was used as the sweep gas on the permeate side. The total duration of all mixed-gas measurements was about 20 days.

## Data availability

The data that support the findings of this study are available in the manuscript, Supplementary Information, and Source Data File. Additional information is also available from the corresponding author upon request. Source data are provided with this paper.

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

## Acknowledgements

This work was funded by the U.S. Department of Energy (DOE) award No. DE-FE0031636 and DE- FE0032209. The authors gratefully acknowledge help from Dr. Xiao Yang for XPS analysis and Dr. Shengwen Liu and Dr. Junjie Chen for Raman analysis. This research used the Materials Synthesis and Characterization and Electron Microscopy Facilities of the Center for Functional Nanomaterials, which is a U.S. DOE Office of Science Facility, at Brookhaven National Laboratory under Contract No. DE-SC0012704.

## Author contributions

All authors contributed to the scientific discussion and manuscript preparation. L.H. and H.L. conceived the approach and conducted experimental designs. L.H. fabricated and characterized materials. W.-I.L., A.S., K.K., S.H., and C.-Y.N. conducted STEM and EELS characterizations and analyses. S.R. and P.A. analyzed BET results. L.Z. measured XPS. S.F. and Y.D. investigated mechanical properties. V.B., T.T., and G.Z. characterized gas separation properties. L.H. wrote the first draft of the manuscript. H.L. supervised the project.

## Competing interests

The authors declare no competing interests.
