## [Peer Review File · Nature Communications]

Hierarchically porous and single Zn atom-embedded carbon molecular sieves for H₂ separationsREVIEWER COMMENTS

Reviewer #1 (Remarks to the Author):

Hu and co-workers described hybrid CMS membranes made from pyrolysis of ZIF-8/PBI mixed-matrix membranes. The hybrid CMS membrane showed improved H₂/CO₂ separation performance over CMS membranes derived from neat PBI under the same pyrolysis parameters. The idea to leverage the chemical similarity between PBI and ZIF-8 was clever. The paper is well written with a large amount of permeation data and characterization data. The H₂/CO₂ separation performance of the sMMM CMS membrane was fine. Overall, the novelty and quality of work are appropriate for Nature Communications and publication may be considered following the revisions noted below.

1. The authors are advised to lighten up the statement that the sMMM CMS membrane had unprecedented performance considering several literature work reported CMS membranes with higher H₂/CO₂ and H₂/hydrocarbon selectivities.
2. ZIF-8 is known to have low intrinsic selectivity for the H₂/CO₂ pair (DOI: 10.1021/jz300855a), so how come the ZIF-8/PBI MMM showed higher H₂/CO₂ selectivity than the PBI matrix (line 73, page 4)? Some discussion would be good.
3. The authors are advised to clarify how many film samples were used to plot the permeation data and add error bars to membrane permeation data if applicable.
4. The work by Hazazi and co-workers (DOI: 10.1016/j.memsci.2022.120548) is quite relevant and a citation may be considered.
5. Did CO₂ permeation have time lags? If yes, they may be added to the SI to show that permeation reached steady state.
6. The reviewer suggests considerable revisions made for Fig. 5e by adding several important literature data (also to Table S8). The authors chose to show both pure gas and mixed gas permeation data of the sMMM CMS membrane in Fig.5e. However, only mixed-gas data are shown for several literature materials. For example, the pure gas H₂/CO₂ permeation data (ideal selectivity over 350) of ref.21 (Iyer and Zhang) may be added to Fig. 5e. Also, results reported in the two papers by Hazazi (DOI: 10.1016/j.memsci.2022.120548) and Iyer (DOI: 10.1016/j.carbon.2023.118598) may be added to Fig. 5e.
7. The authors are advised to remove the CMS upper bound from Fig.5e and the language that sMMM CMS surpasses the CMS upper bound. Unlike polymer upper bounds, there has not been strong theoretical arguments that upper bounds exist for CMS membranes. Second, the drawing of the CMS upper bound seems arbitrary because as mentioned previously, literature data of several high-performing CMS membrane were missing from Fig.5e.
8. What is the rationale to show the 100 degree C polymer upper bound in Fig. 5e considering some literature permeation data shown in the figure were measured at room temperature? How is the 100 degree C polymer upper different from the 2008 Robeson upper bound? A discussion may be added if the authors prefer to keep the 100 degree C polymer upper bound.
9. Dissolving PBI in organic solvents is known to be challenging. The authors are advised to provide more experimental details about how the PBI powders were dissolved in DMF.
10. It was mentioned that the polymer film was sandwiched between two pieces of ceramic disks. The authors are advised to provide more details of the ceramic disks to allow data reproduction, such as the vendor, thickness, and dimension, etc. A schematic or photo will also be useful.
11. Line 68, page 4: the word "method" may be added after "an in-situ growth".
12. Do the authors expect even higher selectivity if the ZIF-8 concentration was increased?

Signed by:
Chen Zhang

Reviewer #2 (Remarks to the Author):

This work reported the development of carbon membranes for H₂ separation, which is an important topic. There are some statements that should be clarified or experimentally approved.

1. In the abstract: "to fine-tune pore bottlenecks of 3.3–3.8 Å", in fact, the authors do not prove

that there are such kinds of ultramicropores.

2. Table S4 listed the solubility of CO₂, C₂H₆, however, the authors use those data to claim that H₂/CO₂ selectivity is dominated by diffusion selectivity. The authors have not obtained the solubility data for H₂, how to make such conclusion?

3. The authors stated that "when carbonized at 400 °C (because of the increased d-spacing) before dramatically increasing (Fig. 3b), suggesting the formation of ultramicropores between 2.89 and 3.8 Å" based on the pure gas permeation testing results (Fig. 3), which has not been well approved (those might be correct for the inert gas such as helium, argon). The authors do not provide the gas sorption data for all the gas molecules (H₂, CO₂, N₂, CH₄). Again, it is not clear whether the molecular sieving mechanism is the only one dominating the gas transport. Based on the pore size distribution given in Fig. 2f and Fig. 4f, the reviewer cannot find there are any pores smaller than 6 Å. If the authors believe that there are ultramicropores (2.9-3.6 Å), we should be able to see the pores of <4Å based on CO₂ adsorption. Actually, carbon membranes usually have a strong adsorption effect, otherwise, there should be not much difference in the selectivity obtained from single gas and mixed gas permeation testing.

4. Fig. S6d gives the N₂ adsorption data, compared to CO₂ adsorption, usually the dV/DW obtained using N₂ adsorption at -196°C will be much higher compared to that from CO₂ adsorption at 0°C, the authors need to explain those results. We also found that the prepared carbon membrane sMMM900 has quite a high micropore volume with a pore size of large 1nm, it is expected that the solubility coefficient may have a significant contribution to gas permeance.

5. Fig. 3f shows that the H₂/CO₂ selectivity decreases with the increase in temperature, please explain it.

6. The authors state that "MMM900 exhibits H₂/CO₂ selectivity of 130 at 70 °C, higher than any known polymeric materials", which does not make sense. The authors should compare with other carbon membranes reported in the literature.

7. The prepared carbon membrane shows very low H₂ permeance (considering a membrane thickness of 35µm, ca. 3GPU), which does not show much competition compared to other carbon membranes reported in the literature.

8. it is difficult to judge the novelty by introducing the MOF particle for carbonization, and how the single-atom metal will contribute to tuning the pore size.

Reviewer #3 (Remarks to the Author):

This work introduces a new concept for the design of high-performance carbon molecular sieve materials (CMS). The materials are based on supramolecular mixed-matrix membranes containing polybenzimidazole and in-situ formed ZIF-8, i.e., sMMMs. Incorporation of ZIF-8 directly embedded in a PBI matrix boosted H₂/CO₂ selectivity (29) compared to pristine PBI (17) or conventional MMMs made by simple physical blending (8.1). The manuscript is well written and easy to follow; the characterizations were expertly performed. Overall, the work is publishable with only minor revisions. I suggest the authors to address the following comments in the revised version of the manuscript:

a) It is commonly known that polymer-derived CMS membranes are brittle, specifically those made at high pyrolysis temperature. I can imagine that pyrolyzed MMMs will show even more pronounced brittle mechanical properties, which will make scale-up of the sMMM concept difficult. In fact, based on the pictures shown in Fig. S1a it appears that sMMMs made at temperatures higher than 550 °C are extremely brittle. Would it be advantageous to reduce the ZIF-8 content in the sMMM to improve the mechanical stability of the membranes without too much loss in gas separation performance?

b) The authors compare the pure- and mixed-gas separation performance of their sMMMs with previously published CMS and polymeric membranes (Fig. 5 e; Table S8). Based on this comparison, the authors claim unprecedented performance for several types of gas separation. For a fair comparison, I suggest the authors to include recently published pure- and mixed-gas data for H₂/CO₂ separation of CMS membranes tested at 35 and 100 °C in Fig. 5 e and Table S8 (Hazazi et al., J. Membr. Sci. 654 (2022) 120548).

RESPONSE TO REVIEWERS' COMMENTS

Reviewer #1:

Hu and co-workers described hybrid CMS membranes made from pyrolysis of ZIF-8/PBI mixed-matrix membranes. The hybrid CMS membrane showed improved H₂/CO₂ separation performance over CMS membranes derived from neat PBI under the same pyrolysis parameters. The idea to leverage the chemical similarity between PBI and ZIF-8 was clever. The paper is well written with a large amount of permeation data and characterization data. The H₂/CO₂ separation performance of the sMMM CMS membrane was fine. Overall, the novelty and quality of work are appropriate for Nature Communications and publication may be considered following the revisions noted below.

We thank positive and valuable comments from the reviewer.

1. The authors are advised to lighten up the statement that the sMMM CMS membrane had unprecedented performance considering several literature work reported CMS membranes with higher H₂/CO₂ and H₂/hydrocarbon selectivities.

We have revised the statements in the manuscript as below:

Page 2, Abstract: “Carbonization temperature is used to fine-tune pore sizes, achieving superior H₂/CO₂, H₂/CH₄, H₂/N₂, and H₂/C₂H₆ separation properties with excellent stability against water vapor and physical aging.”

Conclusions: “Carbonization temperature can be used to engineer hierarchical nano- and micropores including the bottlenecks of 3.3 – 3.8 Å, realizing superior H₂/gas separation performance for the production, transportation, and recovery of H₂ as a clean energy carrier.”

2. ZIF-8 is known to have low intrinsic selectivity for the H₂/CO₂ pair (DOI: 10.1021/jz300855a), so how come the ZIF-8/PBI MMM showed higher H₂/CO₂ selectivity than the PBI matrix (line 73, page 4)? Some discussion would be good.

We have added more discussion and cited the reference on this in the revised manuscript below:

Pages 4-6: “The benzimidazoles in PBI are similar to the ligands (2-methylimidazole or 2-mIm) in forming ZIF-8, and thus, the polymer chains are uniquely incorporated in the ZIF-8, forming amorphous ZIF-8 with strong size-sieving ability in homogeneous MMMs.²⁸ For instance, an sMMM containing amorphous ZIF-8 (11 mass%) and crystalline ZIF-8 (9.1 mass%) was synthesized as the CMS precursor (Table S1), and it exhibits H₂/CO₂ selectivity (29) higher than PBI (17) because of the strong size sieving ability of the amorphous ZIF-8, despite the low H₂/CO₂

selectivity in the crystalline ZIF-8. By contrast, an MMM comprising 10 mass% crystalline ZIF-8 and PBI prepared by physical blending exhibits H₂/CO₂ selectivity of only 8.1.²⁸”

3. The authors are advised to clarify how many film samples were used to plot the permeation data and add error bars to membrane permeation data if applicable.

We tested three films for each CMS sample. We have added error bars for all mixed-gas permeability and selectivity in the revised manuscript as shown in figure 5a-c below. There is a system uncertainty (less than 10%) for all pure-gas permeability. We have added those explanations and updated figures in the revised manuscript as below:

Page 21: “Pure-gas permeability was determined using a constant-volume and variable-pressure apparatus for CMS samples with an effective area of 0.5 - 1 cm². Three samples were tested for each CMS membrane, and all measurements were taken after the steady state was reached. Pure-gas sorption isotherms of CO₂ and C₂H₆ were determined using a gravimetric sorption analyzer of IGA 001 (Hidden Isochema Ltd., Warrington, UK). The uncertainty for gas permeability and solubility is estimated to be <10% using an error propagation method.³⁷”

Page 16: Fig. 5a-c:

4. The work by Hazazi and co-workers (DOI: 10.1016/j.memsci.2022.120548) is quite relevant and a citation may be considered.

We have cited this work as Reference 18.

Page 4: “Pore structures of CMS membranes depend on precursor structures,^{2,5,18} carbonization temperature (T_c),¹⁹ and carbonization atmosphere.^{20,21}”

5. Did CO₂ permeation have time lags? If yes, they may be added to the SI to show that permeation reached steady state.

We did not observe the time lag behavior for CO₂ permeation, and all permeation measurement was conducted until the steady state was reached.

Page 21: “Three samples were tested for each CMS membrane, and all measurements were taken after the steady state was reached.”

6. The reviewer suggests considerable revisions made for Fig. 5e by adding several important literature data (also to Table S8). The authors chose to show both pure gas and mixed gas permeation data of the sMMM CMS membrane in Fig.5e. However, only mixed-gas data are shown for several literature materials. For example, the pure gas H₂/CO₂ permeation data (ideal selectivity over 350) of ref.21 (Iyer and Zhang) may be added to Fig. 5e. Also, results reported in the two papers by Hazazi (DOI: 10.1016/j.memsci.2022.120548) and Iyer (DOI: 10.1016/j.carbon.2023.118598) may be added to Fig. 5e.

We have included the CMS membranes from Hazazi *et al.* (DOI: 10.1016/j.memsci.2022.120548) and Iyer *et al.* (DOI: 10.1016/j.carbon.2023.118598) in Fig. 5e, which are shown as CANAL-TB/900 and MTI aramid/925-35, respectively. Additionally, pure-gas H₂/CO₂ permeation data have been added in Fig. 5e and table S8. The updated Fig.5e is shown below:

Page 16: Fig. 5e

The open and filled symbols indicate mean pure-gas and mixed-gas (H₂/CO₂: 50/50) separation performance, respectively (table S8).

7. The authors are advised to remove the CMS upper bound from Fig.5e and the language that sMMM CMS surpasses the CMS upper bound. Unlike polymer upper bounds, there has not been strong theoretical arguments that upper bounds exist for CMS membranes. Second, the drawing of the CMS upper bound seems arbitrary because as mentioned previously, literature data of several high-performing CMS membrane were missing from Fig.5e.

We have deleted the CMS upper bound from Fig. 5e, and revised the discussion.

Page 17: “Fig. 5e compares H₂/CO₂ separation performance of sMMM900 with state-of-the-art CMS and polymeric materials. Most CMS materials exhibit H₂/CO₂ separation properties superior to polymers and surpassing their upper bound at 100 °C due to the bimodal porous structure. Notably, the upper bound was estimated from Robeson’s one at 35 °C using an activated diffusion model.^{7,33} Nevertheless, the sMMM900 with hierarchical porous structures displays one of the best combinations of H₂ permeability and H₂/CO₂ selectivity among CMS materials.”

Page 16: Fig. 5e

8. What is the rationale to show the 100 degree C polymer upper bound in Fig. 5e considering some literature permeation data shown in the figure were measured at room temperature? How is the 100 degree C polymer upper different from the 2008 Robeson upper bound? A discussion may be added if the authors prefer to keep the 100 degree C polymer upper bound.

We have provided the information requested.

Page 12: “We investigate pure-gas H₂/CO₂ separation behaviors of sMMM, sMMM550, and sMMM900 at 70 – 150 °C, an interesting temperature range for syngas processing.⁷”

Page 17: “Most CMS materials exhibit H₂/CO₂ separation properties superior to polymers and surpassing their upper bound at 100 °C due to the bimodal porous structure. Notably, the upper bound was estimated from Robeson’s one at 35 °C using an activated diffusion model.^{7,33}”

9. Dissolving PBI in organic solvents is known to be challenging. The authors are advised to provide more experimental details about how the PBI powders were dissolved in DMF.

We have added more details on the preparation of the PBI/DMF solution in the revised manuscript as shown below:

Page 18: “First, 10 g PBI powders were added to 90 g DMF, and the solution was heated at 160 °C for 12 h to allow the PBI to dissolve. After being filtered through a 1- μ m glass fiber filter, a solution with ~ 6 wt% PBI concentration was obtained and then diluted to 3 wt% by adding more DMF.”

10. *It was mentioned that the polymer film was sandwiched between two pieces of ceramic disks. The authors are advised to provide more details of the ceramic disks to allow data reproduction, such as the vendor, thickness, and dimension, etc. A schematic or photo will also be useful.*

We have provided more details and a schematic in the manuscript and SI as below:

Page 19: “To prepare a CMS film, an sMMM or PBI film (with a mass of m_0) was sandwiched between two nonporous alumina ceramic sheets (4.5 inches \times 4.5 inches \times 0.025 inches, McMaster Carr, IL). The assembly was placed in a tube furnace with a N₂ flow of 200 mL/min (fig. S10).”

Page S21, Fig S10, SI:

11. *Line 68, page 4: the word “method” may be added after “an in-situ growth”.*

We have added in the revised manuscript as below:

Page 4: “Here, we report a distinct series of hybrid CMS materials derived from a supramolecular MMM (sMMM) containing zeolitic imidazolate framework-8 (ZIF-8) in polybenzimidazole (PBI) synthesized by an in-situ growth method.”

12. *Do the authors expect even higher selectivity if the ZIF-8 concentration was increased?*

The reviewer raised a very good question. We agree that the effect of the ZIF-8 concentration can significantly influence the selectivity. However, this can only be answered by more in-depth studies. We have added the following sentences.

Page 18: “We expect that the sMMM CMS films can be further optimized using a variety of parameters to achieve a desirable combination of separation properties and mechanical strengths, including ZIF-8 concentration, MOF types, carbonization temperature, etc. Future work will also need to develop these materials into industrial membranes with thin selective layers to obtain high gas permeance.”

Reviewer #2:

This work reported the development of carbon membranes for H₂ separation, which is an important topic. There are some statements that should be clarified or experimentally approved.

1. In the abstract: “to fine-tune pore bottlenecks of 3.3–3.8 Å”, in fact, the authors do not prove that there are such kinds of ultramicropores.

We have revised the manuscript to avoid the confusion.

Page 2: “Carbonization temperature is used to fine-tune pore sizes, achieving superior H₂/CO₂, H₂/CH₄, H₂/N₂, and H₂/C₂H₆ separation properties with excellent stability against water vapor and physical aging.”

2. Table S4 listed the solubility of CO₂, C₂H₆, however, the authors use those data to claim that H₂/CO₂ selectivity is dominated by diffusion selectivity. The authors have not obtained the solubility data for H₂, how to make such conclusion?

We have provided the information to clarify the confusion. Additionally, a few reports have shown H₂/CO₂ solubility of ~0.03 in polymers and liquids (Hu et al., Molecularly Engineering Polymeric Membranes for H₂/CO₂ Separation at 100–300 °C. *J. Polym. Sci.* 58 (2020) 2467-2481, DOI 10.1002/pol.20200220). Therefore, the high H₂/CO₂ selectivity (up to 130) in these CMS materials is derived from the high diffusivity selectivity.

Page 3: “H₂ is less condensible than other gases, as indicated by its lower critical temperature, and thus, it has unfavorable solubility selectivity. Therefore, for H₂ separations, membrane materials should have a strong size-sieving ability because H₂ (with a kinetic diameter of 2.89 Å) is smaller than other gases, such as CO₂ (3.3 Å), N₂ (3.64 Å), CH₄ (3.8 Å), and C₂H₆ (4.44 Å).”

Page 12: “Due to its low condensability, H₂ sorption is below the detection limit of our sorption apparatus, and C₂H₆ can be used as a surrogate for H₂.³³”

3. The authors stated that “when carbonized at 400 °C (because of the increased d-spacing) before dramatically increasing (Fig. 3b), suggesting the formation of ultramicropores between 2.89 and 3.8 Å” based on the pure gas permeation testing results (Fig. 3), which has not been well approved

(those might be correct for the inert gas such as helium, argon). The authors do not provide the gas sorption data for all the gas molecules (H₂, CO₂, N₂, CH₄). Again, it is not clear whether the molecular sieving mechanism is the only one dominating the gas transport. Based on the pore size distribution given in Fig. 2f and Fig. 4f, the reviewer cannot find there are any pores smaller than 6 Å. If the authors believe that there are ultramicropores (2.9-3.6 Å), we should be able to see the pores of <4Å based on CO₂ adsorption. Actually, carbon membranes usually have a strong adsorption effect, otherwise, there should be not much difference in the selectivity obtained from single gas and mixed gas permeation testing.

The reviewer raised very important questions. First, we would like to confirm that gas sorption is an important step of its transport, as it is described using the solution-diffusion model. Notably, the high H₂/CO₂ selectivity in these CMS materials is derived from their strong size-sieving ability and thus high diffusivity selectivity, as H₂/CO₂ solubility selectivity is always less than 1.

Page 12: “Gas transport in CMS materials is usually described using the solution-diffusion model, where permeability is decoupled into gas solubility and diffusivity.^{5,18} CO₂ and C₂H₆ sorption isotherms were determined at 100 °C (fig. S8). Due to its low condensability, H₂ sorption is below the detection limit of our sorption apparatus, and C₂H₆ can be used as a surrogate for H₂.³³”

Second, our argument on the sizes of the ultramicropores is based on extremely high selectivity values, such as 130 for H₂/CO₂ selectivity. Therefore, we expect that there are ultramicropores smaller than the kinetic diameter of CO₂ (3.3 Å). However, such small pores cannot be detected by either CO₂ sorption (0 °C) or N₂ sorption (−196 °C). We have added these sorption isotherms in Fig. S6b-e of SI. Notably, high-pressure CO₂ was used.

Page 10: “Both CO₂ (0 °C) and N₂ sorption (−196 °C) lead to similar values of pore volumes and surface areas, which has been observed for carbon materials with carbonization-derived mass losses of 5 – 30%.³¹ Notably, both sorption measurements do not show ultramicropores of 3-5 Å because N₂ is too large to access the ultramicropores,^{5,20} and our apparatus cannot obtain stable CO₂ pressure ranges below 10⁻⁵ to accurately evaluate ultramicropores.”

Fig.S6 b-f, SI:

Fig.S7, SI:

Fig. S8. (a) CO₂ and (b) C₂H₆ sorption isotherms of sMMM and sMMM CMS at 100 °C. (c) CO₂ and (d) C₂H₆ sorption isotherms of MMM550 and PBI550 at 100 °C.

4. Fig. S6d gives the N₂ adsorption data, compared to CO₂ adsorption, usually the dV/DW obtained using N₂ adsorption at -196°C will be much higher compared to that from CO₂ adsorption at 0°C, the authors need to explain those results. We also found that the prepared carbon membrane sMMM900 has quite a high micropore volume with a pore size of large 1nm, it is expected that the solubility coefficient may have a significant contribution to gas permeance.

First, we would like to reiterate that gas sorption is an important step of its transport and contributes to permeability, as it is described using the solution-diffusion model. Notably, the high H₂/CO₂ selectivity in these CMS materials is derived from their strong size-sieving ability and thus high diffusivity selectivity, as solubility selectivity is always less than 1.

Page 12: “Gas transport in CMS materials is usually described using the solution-diffusion model, where permeability is decoupled into gas solubility and diffusivity.^{5,17}”

Second, both CO₂ sorption (0 °C) and N₂ sorption (−196 °C) have been used to characterize activated carbon. It was reported at low mass loss values (<5%), micropore volumes from N₂ measurement are less than that from CO₂ measurement; at medium mass loss values (<35%), both measurements yield similar micropore volumes; and at high mass loss values, micropore volumes from N₂ measurement is more than that from CO₂ measurement (J. Garrido, A. Linares-Solano, J. M. Martin-Martinez, M. Molina-Sabio, F. Rodriguez-Reinoso, R. Torregrosa. Use of nitrogen vs. carbon dioxide in the characterization of activated carbons. *Langmuir*. 3 (1987) 76-81, 10.1021/la00073a013). The CMS materials in this study have medium mass losses (20 – 35%), and therefore, the values from both measurements are fairly close. We also revise the manuscript to clarify this issue.

Page 10: “Both CO₂ (0 °C) and N₂ sorption (−196 °C) lead to similar values of pore volumes and surface areas, which has been observed for carbon materials with carbonization-derived mass losses of 5 – 30%.³¹ Notably, both sorption measurements do not show ultramicropores of 3-5 Å because N₂ is too large to access the ultramicropores,^{5,20} and our apparatus cannot obtain stable CO₂ pressure ranges below 10⁻⁵ to accurately evaluate ultramicropores.”

5. *Fig. 3f shows that the H₂/CO₂ selectivity decreases with the increase in temperature, please explain it.*

We have added the explanations on this in the revised manuscript as below:

Pages 12-13: “Both H₂ and CO₂ permeability increases with increasing temperature, and they can be satisfactorily described using the Arrhenius equation (Fig. 3e).³⁴ Compared to sMMM and sMMM900, sMMM550 has the lowest value of activation energy ($E_{P,A}$) for H₂ and CO₂ permeation (tables S6,7) because of the combination of high porosity and large ultramicropores; sMMM900 exhibits the highest $E_{P,A}$ value for H₂ and CO₂ permeation among all samples (table S7) because of its lowest d -spacing and smallest ultramicropores. Interestingly, sMMM900 exhibits the most negative value of ($E_{P,H_2} - E_{P,CO_2}$) among all sMMM CMS samples, leading to a decreased H₂/CO₂ selectivity with increasing temperature (Fig. 3f). This behavior is consistent with the strongest size-sieving ability in sMMM900. Similarly, MMM900 exhibits H₂/CO₂ selectivity decreasing with increasing temperature and a negative value of ($E_{P,H_2} - E_{P,CO_2}$), while MMM and MMM550 display the selectivity independent of temperature because of the weak size-sieving ability (fig. S9).²”

6. The authors state that “MMM900 exhibits H₂/CO₂ selectivity of 130 at 70 °C, higher than any known polymeric materials”, which does not make sense. The authors should compare with other carbon membranes reported in the literature.

We have revised the statement to clarify the confusion.

Page 13: “Noticeably, sMMM900 exhibits H₂/CO₂ selectivity of 130 at 70 °C, the highest value obtained for our CMS materials in this study.”

7. The prepared carbon membrane shows very low H₂ permeance (considering a membrane thickness of 35 μm, ca. 3 GPU), which does not show much competition compared to other carbon membranes reported in the literature.

We agree with the reviewer that the freestanding films prepared have low permeance due to the high thickness (35 μm). Nevertheless, this study focuses on fundamental understandings of the intrinsic structure/property relationship of sMMM CMS. High permeance can be achieved if these materials can be fabricated into industrial membranes with a selective layer thickness of 1 μm or less. We have revised the manuscript to clarify the confusion.

Pages 17-18: “Future work will also need to develop these materials into industrial membranes with thin selective layers to obtain high gas permeance.”

8. it is difficult to judge the novelty by introducing the MOF particle for carbonization, and how the single-atom metal will contribute to tuning the pore size.

We agree with the reviewer that it is difficult to assess directly how single metal atoms influence the nanopore sizes, but we respectfully disagree with the reviewer on the novelty of the work. We believe that this is the first effort to demonstrate that single metal atoms improve H₂/CO₂ separation properties. Instead of reiterating the data and discussion from the manuscript, we have included the comments from the other two reviewers on the novelty of this work.

Reviewer 1: “The hybrid CMS membrane showed improved H₂/CO₂ separation performance over CMS membranes derived from neat PBI under the same pyrolysis parameters. **The idea to leverage the chemical similarity between PBI and ZIF-8 was clever.** Overall, the novelty and quality of work are appropriate for Nature Communications and publication may be considered following the revisions noted below.”

Reviewer 3: “This work introduces **a new concept** for the design of high-performance carbon molecular sieve materials (CMS). The materials are based on supramolecular mixed-matrix membranes containing polybenzimidazole and in-situ formed ZIF-8, i.e., sMMMs.”

Reviewer #3:

This work introduces a new concept for the design of high-performance carbon molecular sieve materials (CMS). The materials are based on supramolecular mixed-matrix membranes containing polybenzimidazole and in-situ formed ZIF-8, i.e., sMMMs. Incorporation of ZIF-8 directly embedded in a PBI matrix boosted H₂/CO₂ selectivity (29) compared to pristine PBI (17) or conventional MMMs made by simple physical blending (8.1). The manuscript is well written and easy to follow; the characterizations were expertly performed. Overall, the work is publishable with only minor revisions. I suggest the authors to address the following comments in the revised version of the manuscript:

We thank the reviewer for the positive feedback.

1. It is commonly known that polymer-derived CMS membranes are brittle, specifically those made at high pyrolysis temperature. I can imagine that pyrolyzed MMMs will show even more pronounced brittle mechanical properties, which will make scale-up of the sMMM concept difficult. In fact, based on the pictures shown in Fig. S1a it appears that sMMMs made at temperatures higher than 550 °C are extremely brittle. Would it be advantageous to reduce the ZIF-8 content in the sMMM to improve the mechanical stability of the membranes without too much loss in gas separation performance?

We agree with the reviewer that the ZIF-8 loading and carbonization temperature may be decreased to improve mechanical properties of the CMS films. However, the improved mechanical properties need to be considered with the potentially reduced separation properties. Additionally, these materials will likely be fabricated on porous supports, and thus, the optimization will be more relevant during the fabrication of industrial membranes. We have included the following sentences

Pages 17-18: “We expect that the sMMM CMS films can be further optimized using a variety of parameters to achieve a desirable combination of separation properties and mechanical strengths, including ZIF-8 concentration, MOF types, carbonization temperature, etc. Future work will also need to develop these materials into industrial membranes with thin selective layers to obtain high gas permeance.”

2. The authors compare the pure- and mixed-gas separation performance of their sMMMs with previously published CMS and polymeric membranes (Fig. 5e; Table S8). Based on this comparison, the authors claim unprecedented performance for several types of gas separation. For a fair comparison, I suggest the authors to include recently published pure- and mixed-gas data for H₂/CO₂ separation of CMS membranes tested at 35 and 100 °C in Fig. 5 e and Table S8 (Hazazi et al., J. Membr. Sci. 654 (2022) 120548).

We have included the CMS membranes from *Hazazi et al.* (DOI: 10.1016/j.memsci.2022.120548) in Fig. 5e, which is shown as CANAL-TB/900 (symbols of 13 and 14). We also add another recently reported CMS membrane (10.1016/j.carbon.2023.118598), which is shown as MTI aramid/925-35 (symbol of 11). The updated Fig.5e is shown below:

Page 16: Fig. 5e

REVIEWERS' COMMENTS

Reviewer #1 (Remarks to the Author):

The authors have adequately addressed reviewer comments. The revised manuscript is now in a good shape for acceptance.

Signed by Chen Zhang

Reviewer #2 (Remarks to the Author):

the authors made a great effort to address the comments raised by reviewers, and I recommend accepting the current version for publishing.

Reviewer #3 (Remarks to the Author):

The authors properly addressed essentially all major comments of the reviewers in the revised manuscript. The quality of the work has now been raised to the appropriate level for publication in Nature Communications. NO additional modifications are necessary.